# GenoVi, an open-source automated circular genome visualizer for bacteria and archaea

**Andrés Cumsille**[1], **Roberto E. Durán**[1]*, **Andrea Rodríguez-Delherbe**[2,3], **Vicente Saona-Urmeneta**[3], **Beatriz Cámara**[1], **Michael Seeger**[1], **Mauricio Araya**[4], **Nicolás Jara**[4], **Carlos Buil-Aranda**[3]

**1** Laboratorio de Microbiología Molecular y Biotecnología Ambiental, Departamento de Química & Centro de Biotecnología Daniel Alkalay Lowitt, Universidad Técnica Federico Santa María, Valparaíso, Chile, **2** Radcliffe Department of Medicine, MRC Weatherall Institute of Molecular Medicine, University of Oxford, Oxford, United Kingdom, **3** Departamento de Informática, Universidad Técnica Federico Santa María, Valparaíso, Chile, **4** Departamento de Electrónica, Universidad Técnica Federico Santa María, Valparaíso, Chile

☉ These authors contributed equally to this work.

* roberto.duranv@usm.cl

**Data Availability Statement:** GenoVi is freely available under a BY-NC-SA Creative Commons License and can be downloaded from https://github.com/robotoD/GenoVi. GenoVi can be obtained in two steps: Creating a Conda

## Abstract

The increase in microbial sequenced genomes from pure cultures and metagenomic samples reflects the current attainability of whole-genome and shotgun sequencing methods. However, software for genome visualization still lacks automation, integration of different analyses, and customizable options for non-experienced users. In this study, we introduce GenoVi, a Python command-line tool able to create custom circular genome representations for the analysis and visualization of microbial genomes and sequence elements. It is designed to work with complete or draft genomes, featuring customizable options including 25 different built-in color palettes (including 5 color-blind safe palettes), text formatting options, and automatic scaling for complete genomes or sequence elements with more than one replicon/sequence. Using a Genbank format file as the input file or multiple files within a directory, GenoVi (i) visualizes genomic features from the GenBank annotation file, (ii) integrates a Cluster of Orthologs Group (COG) categories analysis using DeepNOG, (iii) automatically scales the visualization of each replicon of complete genomes or multiple sequence elements, (iv) and generates COG histograms, COG frequency heatmaps and output tables including general stats of each replicon or contig processed. GenoVi's potential was assessed by analyzing single and multiple genomes of *Bacteria* and *Archaea*. *Paraburkholderia* genomes were analyzed to obtain a fast classification of replicons in large multipartite genomes. GenoVi works as an easy-to-use command-line tool and provides customizable options to automatically generate genomic maps for scientific publications, educational resources, and outreach activities. GenoVi is freely available and can be downloaded from https://github.com/robotoD/GenoVi.

environment with Circos, followed by installation using the package-management system pip with pip install genovi. Also, a Docker container of GenoVi is available. Genomes used in this study are available at https://zenodo.org/record/7331473.

**Funding:** This work was supported by USM_PI_M_43 Proyecto USM Multidisciplinarios 2020, - Universidad Técnica Federico Santa María (UTFSM; A.C., R.E.D., V.S.-U., M.A., N.J., C.B.-A.), Fondecyt 1200756 - Agencia Nacional de Investigación y Desarrollo (ANID; M.S., R.E.D.), and ANID – Millennium Science Initiative Program – Code ICN17_002 (C.B.-A.) grants. A.C. was supported by ANID 21191625 PhD fellowship and Programa de Incentivos a la Iniciación Científica, UTFSM. The funders had no role in software design, data collection or analysis, decision to publish or the preparation of the manuscript.

**Competing interests:** The authors have declared that no competing interests exist.

## Author summary

Genome visualization tools can inspect genomic features in a DNA sequence, delivering a visual aid to quickly understand genome architecture and function. Circular representations frequently display the GC content, useful to identify genomic islands and horizontal gene transfer events; GC skew, the over or under abundance of G or C between the leading and lagging DNA strands frequently used to identify the origin and terminus of replication; coding DNA sequences (CDS), and Clusters of Orthologous Groups (COGs) to classify predicted CDS for functional studies. However, genome visualization tools frequently require these features in specific formatting as input, hampering their usage, and lacking versatility for comparative genomics purposes. GenoVi uses an annotated genome file as input, automatically calculates each of the aforementioned genomic features, and generates a ready-to-use figure in minutes. Additionally, GenoVi has many customizable format options and works with complete, draft, and multiple genomes useful for comparative genomics applications.

This is a *PLOS Computational Biology* Software paper.

## Introduction

The growth of genomic data has resulted in more than 250,000 unique bacterial and archaeal genomes available in public databases since the first genome was sequenced [1,2]. Large-scale projects, international research collaborations, and smaller groups employ genomics and environmental genomics to pursue new knowledge and understand complex questions in evolution, ecology, systematics, biomedical sciences, and other areas, generating extensive amounts of data while discovering new patterns and mechanisms within life sciences [3]. Metagenome-assembled genomes (MAGs), which are increasing day-by-day due to metagenomic studies [1], as well as the current genome sequencing accessibility for most research groups, build up the need to create automated and easy-to-use tools to analyze and interpret its information.

Circular genome visualization is a widely used method for data analysis and representation of genomic elements. General features usually displayed in a circular representation includes the GC content, the guanine and cytosine uneven proportion in the two DNA strands, phenomenon called GC skew [4], classification of proteins into Clusters of Orthologous Groups of proteins (COGs) [5,6], and the location of tRNAs and rRNAs in the genome. Different tools are available for circular genome visualization, such as CGView [7], CiVi [8], DNAPlotter [9], Circleator [10], and Circos [11]. However, their usage often accepts only complete chromosomes or plasmids for circular representation (Table 1). Others require programming skills for the creation of complex intermediate and configuration files increasing the gap between users and graphical visualization. Complementary analyses for genome visualization, such as COGs classification and configuration files, as well as custom colors, are not easy to achieve, hampering their implementation in visualization tools (Table 1) [3].

This work presents GenoVi, a Python-based tool that automatically formats each file to create a circular representation of bacterial or archaeal genomes integrating multiple tools. GenoVi displays COGs annotation, coding DNA sequences (CDS), GC content, GC skew, tRNA, and rRNA localization using a GenBank format file (gbff, gbk, gb) as an input file. This tool

**Table 1. Comparison of circular genome visualization software.**

| | Genomic features | | | | | File input | | Customization | | Interface | Comments | Reference |
|---|---|---|---|---|---|---|---|---|---|---|---|---|
| | GC content | GC skew | CDS | RNA | COG | Single scaffold | Multiple scaffold | Multiple replicons | Prebuilt Palettes | | | |
| **GenoVi** | + | + | + | + | + | + | + | + | + | Command line | Add all selected features in one step | This study |
| **CGView** | + | + | + | + | - | + | - | - | - | Command line | COGs or other features can be added using a gff file | [7] |
| **Proksee** | + | + | + | + | - | + | + | - | - | Interactive web platform | Customizable colors and size. No difference between draft or complete genomes. Integration with annotation software. | https://proksee.ca/ |
| **CiVi** | + | + | + | + | - | + | - | - | - | Web | Unavailable. COGs or other features can be added manually. | [8] |
| **DNAPlotter** | + | + | + | + | - | + | - | - | - | Interactive Java application | Extra features can be added manually. | [9] |
| **Circleator** | + | + | + | + | - | + | - | - | - | Command line | Each feature should be upload independently | [10] |
| **Circos** | - | - | - | - | - | + | + | - | - | Command line | Genome features can be added manually | [11] |

bypasses the difficulties associated with data processing, specific formatting such as the configuration files required by Circos and several customization options, delivering a high-quality figure using complete or draft genomes. Additionally, GenoVi creates output files with COGs classification information in minutes, as well as general features tables, making it useful for single genome representation, and comparative genomic studies.

## Design and implementation

GenoVi is a command-line tool that compiles different software to create a ready-to-publish circular genome representation. GenoVi automatically calculates the GC content and GC skew from a genome, and unless specified, assigns CDS to COG categories. As additional resources, GenoVi produces histograms, heatmaps and tables of COG categories and frequency, and a table with general information about each contig/replicon, including size, GC content, number of CDS, tRNAs, and rRNAs. The originality of GenoVi resides in being a one-step, and easy-to-use tool that computes all the information needed to create a customizable genome representation in a matter of minutes, using as input an annotated genome. GenoVi can be used for single genome visualization or for comparative genomic studies.

## GenoVi workflow

As input, GenoVi uses a Genbank format file (gbff), which is converted into a nucleotide fasta (.fna) using a modified version of genbank2fasta tool. Then, the fasta file is used to calculate the GC content using a script based on GC-analysis.py [12], and the GC skew using SkewIT for any sequence element identified in the input file [4]. In both cases, GenoVi uses a user-specified window size for calculation (default = 1,000 bp). Concurrently, from the original gbff file, GenoVi obtains the position of CDS, tRNA, and rRNA associated with each sequence element.

As a default feature, GenoVi parses the gbff file into a protein fasta (.faa) to predict and classify each CDS into COG categories using the COG 2020 database. This feature is accomplished using DeepNOG, a fast alignment-free method based on convolutional network architecture achieved within a few minutes. [13]. After calculating each genomic feature, GenoVi creates configuration files needed to display a circular representation using Circos. In addition, GenoVi delivers as output a histogram of COG categories abundance, a heatmap of COG frequency per contig/replicon, and three output tables: COG classification raw data, COG percentage distribution per genome/replicon, and general features which could be used for further analyses.

## Usage

GenoVi is a Python-based command-line software installed by creating a Circos [11] containing Conda environment [14]. GenoVi can then be installed through pip, which also incorporates DeepNOG [13], and Python libraries (NumPy, Pandas, Biopython, Matlibplot, and CairoSVG). GenoVi can also be installed from our git repository, previously installing each dependency.

GenoVi relies upon user indication of the input file and the genome status ("draft" or "complete"; Fig 1A). For our purposes, draft genomes are DNA assemblies fragmented in contigs or scaffolds interspersed with gaps of unknown length, whereas complete genomes have defined length gaps filled with "N" (any nucleotide) or no gaps, in which each scaffold represents an

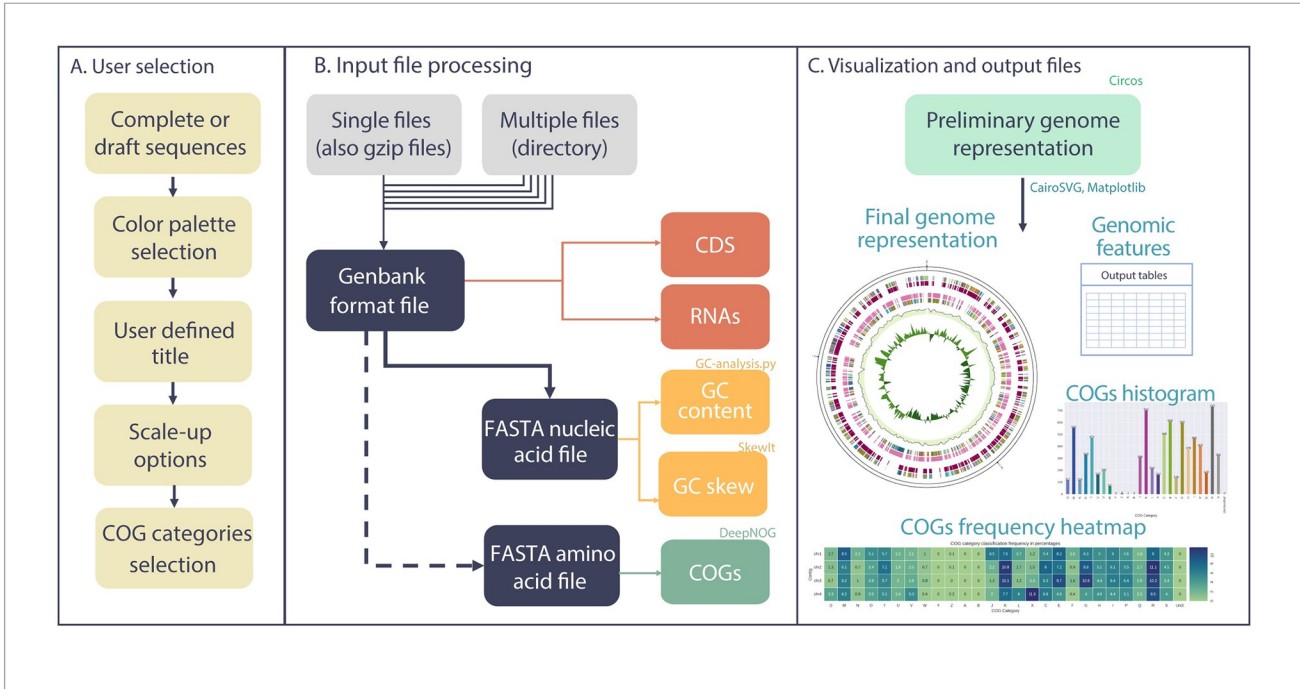

**Fig 1. Overview of GenoVi workflow.** (A) GenoVi incorporates several customizable options for visualization, such as an option for complete or draft genomes, twenty-five prebuilt color palettes, title, scale-up options, and COG categories selection. (B) GenoVi uses Genbank format files (gbff) as input. From each file GenoVi extracts CDS and RNAs position. Additionally, GenoVi converts this file into a nucleotide FASTA to calculate GC content, GC skew, and, if user-specified, into a protein FASTA to classify into COGs categories. (C) After calculating and formatting each genomic feature, GenoVi uses Circos to build a genome representation in svg and png format. COG abundance histogram, COG frequency heatmaps and summarizing tables of overall genomic features are also created in this step.

independent replicon (*e.g.* chromosome, chromid, megaplasmid or plasmid). We encourage to use complete genomes with N-filled gaps assemblies only when necessary, due to the inability to obtain genomic features from that data. The genome status argument defines two main methods for visualization: *-draft*, incorporating each scaffold as bands in the same unique circular representation; and *-complete* where each scaffold is treated separately to generate a circular representation. Genovi can also use directories containing several genomes as an input, treating each genome individually as draft or complete depending on the user selection, useful for comparative genomic analysis.

GenoVi can be used with genbank format files from official annotators of the International Nucleotide Sequence Database Collaboration (Fig 1B), including the NCBI Prokaryotic Genome Annotation Pipeline from GenBank [15] and the DDBJ Fast Annotation and Submission Tool (DFAST) from the DNA Database of Japan (DDBJ; [16]), as well as annotation files from Prokka [17].

GenoVi has several user-customizable options. Color selection for CDS, GC content, GC skew, tRNA, rRNA, font, and background. Additionally, GenoVi includes twenty-five pre-built color palettes suitable for visual representation of microbial genomes from different environments or contexts, including five color-blind friendly palettes. A figure title and replicon size display option are available. Additionally, italicized words in the title can be added for precise taxonomic nomenclature. For complete genomes with more than one replicon, e.g., one chromosome and plasmids, three scaling options (viz. variable, linear, sqrt) are offered. This option is especially useful for genome containing broad differences in replicon size which could affect the illustration of small sequence elements (Fig 1A).

To enhance and customize the genomic feature estimations, the user can specify the window size in bp for GC content and GC skew calculations, and the confidence threshold for DeepNOG COGs classification. To avoid re-processing when deciding image preferences, two options are available: keep temporary files (-k), and reuse COG annotation performed by DeepNOG (-r). COGs categories can be selectively displayed using the—*cogs* argument, choosing specific categories (—*cogs* EMX), a group of COG categories (—*cogs inf-*, for Information Storage and Processing: ABJKLX) or the N-most abundant COGs identified (—*cogs* #, representing a numerical value of top categories to be displayed). Genomic maps created by GenoVi are obtained as PNG and SVG files (Fig 1C), which could be further edited in vector files editing programs such as Adobe Illustrator, Inkscape, Vector, or Microsoft PowerPoint.

GenoVi outputs tables and visualizations are suitable to be used as single genome representation figures, as well as for comparative genomic studies.

## Results and discussion

GenoVi can be used to visualize and analyze data obtained from (i) draft genomes, (ii) complete genomes, (iii) and multiple genomes, converting it into a suitable tool for single and comparative genomics.

To perform genome visualization and analysis of a genomic sequence, a keyword feature is available to indicate whether the DNA is completely sequenced or in draft status. For draft genomes (Fig 2A), GenoVi creates one map that includes each scaffold or contig in the same circular plot. The genome of the type strain of *Corynebacterium alimapuense* VA37-3$^{T}$ [Accession Number: GCF_003716585.1; 18] was used as an example. Genome assembly yielded 12 contigs, where five hold most of the genomic information (Fig 2A). The total assembly length is 2.3 Mb, and has a GC content of 57%, standing slightly above the average of the genus (~55%) [4]. The most representative COG categories in strain VA37-3$^{T}$ are J (translation, ribosomal structure, and biogenesis), E (amino acid transport and metabolism), and R (general

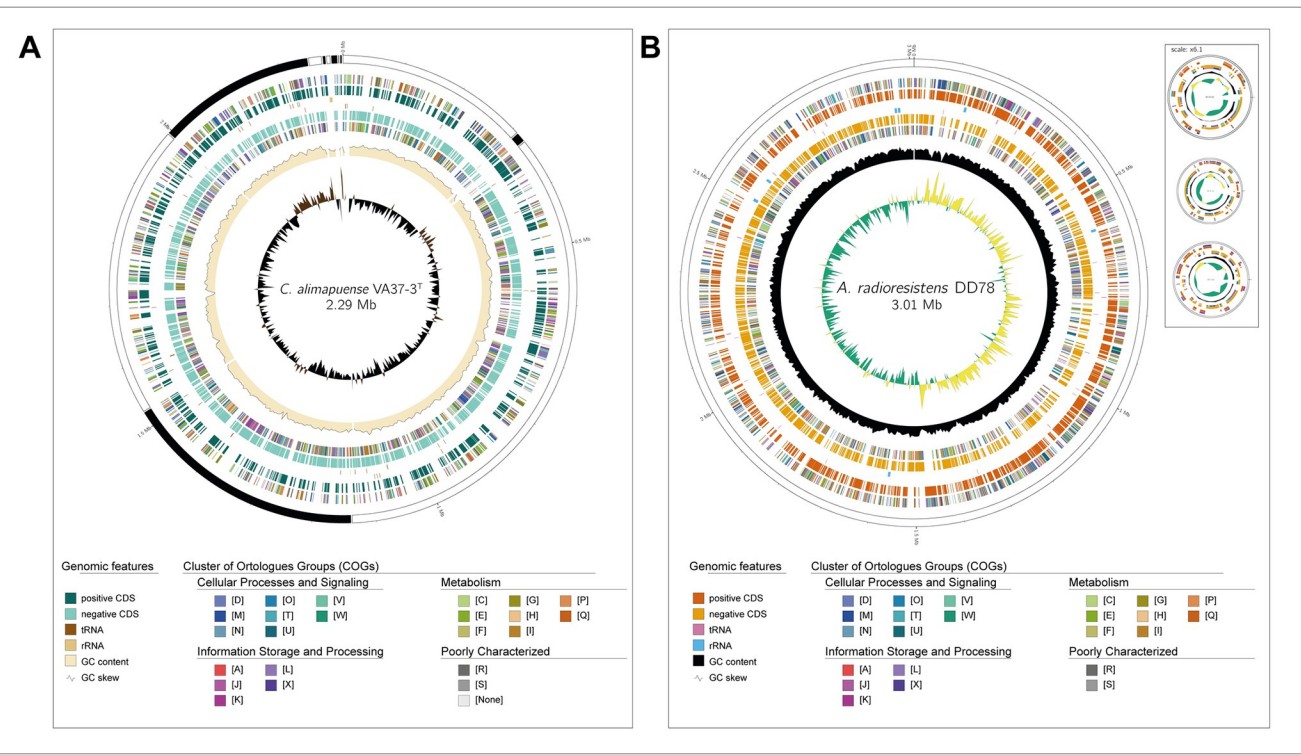

**Fig 2. Application of GenoVi using a draft or complete bacterial genome.** (A) GenoVi circular map of the draft genome of *Corynebacterium alimapuense* VA37-3[T] [18] (-s *draft -cs paradise*). Each contig is represented as separated bands of one circular representation. (B) GenoVi circular representation of the complete genome of *Acinetobacter radioresistens* DD78 [21] (-s *complete, -cs autumn*). Each circular map represents a replicon from the complete genome. *A. radioresistens* DD78 genome consists of a circular chromosome and three circular plasmids displayed next to the chromosome (—*scale* variable). Labeling from outside to the inside: Contigs; COGs on the forward strand; CDS, tRNAs, and rRNAs on the forward strand; CDS, tRNAs, and rRNAs on the reverse strand; COGs on the reverse strand; GC content; GC skew.

function prediction only). Similar results were obtained in *Corynebacterium diphtheriae* strains, where E and J COG categories were the most abundant [19].

GC skew is the guanine-cytosine asymmetry observed when comparing the leading and lagging DNA strands in a continuous sequence. Graphical representation or cumulative GC skew plots shows an inflection point used to identify the origin and terminus (*ori/ter*) of replication in bacteria. Also, mean GC skew values tend to be rather similar across bacterial genera, being a good method to identify misassemblies [4,20]. Due to the draft nature of strain VA37-3[T] assembly, GC skew only reveals which contigs harbor a possible *ori/ter* (Fig 2A).

For complete genome visualization, GenoVi creates a circular map for each replicon in the input file, assuming that each continuous sequence is independent. GenoVi scales each representation according to its length, providing the option to choose different scaling algorithms. *Acinetobacter radioresistens* DD78 [Accession Number: GCF_005519305.1; 21] (Fig 2B) was used as a complete genome example with multiple replicons. The chromosome is 3.0 Mb and has a GC content of 41.8%. In contrast, the three plasmid sizes are 88.5 kb, 80.1 kb, and 69.2 kb with a GC content of 38.9%, 40.7%, and 37.1%, respectively. The overall GC content is 41.6% which is above the average of the genus of 39.4%. The GC skew of strain DD78 shows two inflection points where possibly the origin and the terminus are located (Fig 2B) [4]. The most abundant COG categories were R, J, and M (cell wall/membrane/envelope biogenesis). These results are distant from those reported for *Acinetobacter venetianus* VE-C3, which

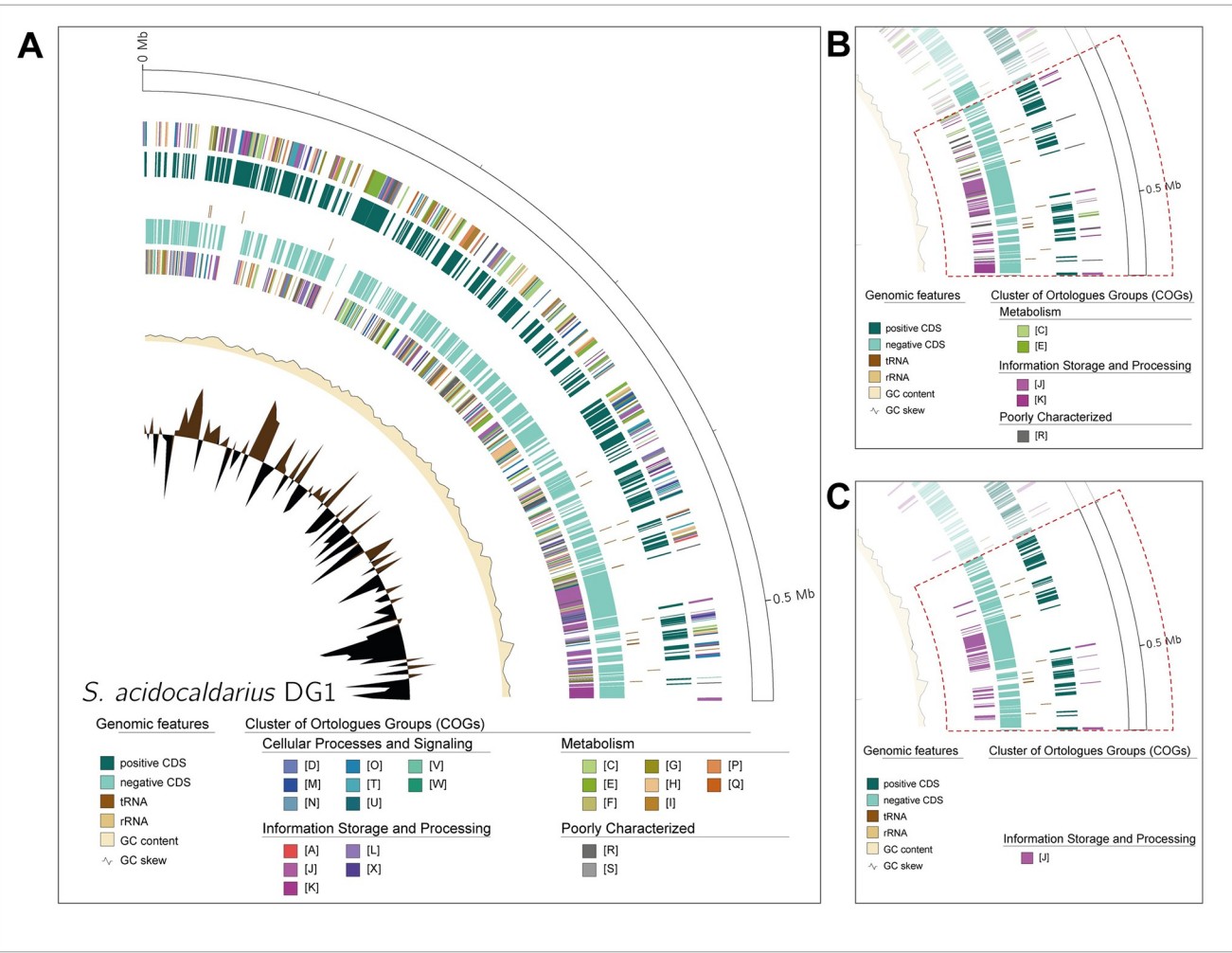

**Fig 3. High density of Translation, ribosomal structure and biogenesis (J) orthologs is located in the region 400–500 kb of *Sulfolobus acidocaldarius* DG1 genome.** (A) Genovi representation of the complete genome of *S. acidocaldarius* DG1 is depicted using default parameters (*-s complete -cs paradise*). (B) Most abundant five COG categories within the 400–550 kb range of DG1 genome are represented (—*cogs 5*), which included C (Energy production and conversion), E (Amino acid transport and metabolism), J (Translation, ribosomal structure, and biogenesis), K (Transcription) and R (General function prediction only). (C) Genovi representation of J category orthologs within the 400–550 kb genome of *S. acidocaldarius* DG1 (—*cogs J*).

reports a majority in the L (replication, recombination, and repair) COG category [22]. Strain DD78 possesses 77 tRNAs and 21 rRNAs, all located at the chromosome (Fig 2B).

For a comprehensive display of COG categories, GenoVi holds several options for COG representation. Otherwise stated using the—*cogs* argument, every COG category will be illustrated (Figs 2 and 3A). The complete genome of the *Crenarchaeota*, *Sulfolobus acidocaldarius* DG1 (GCA_002215565.1, *-s complete -cs paradise*) is illustrated (Fig 3A). Selecting only the most abundant 5 COGs within the DG1 genome, we can observe a high density of J and K (Replication) orthologs in the region between 400 and 550 kb (—*cogs 5*; Fig 3B). When only displaying CDS classified as J orthologs, we can observe that most genes within that range encode for proteins related to translation, ribosomal structure, and biogenesis processes (—*cogs J*; Fig 3C). Selective COG display as seen in the genome of strain DG1, allows the analysis and location of specific functional categories within each genome depicted.

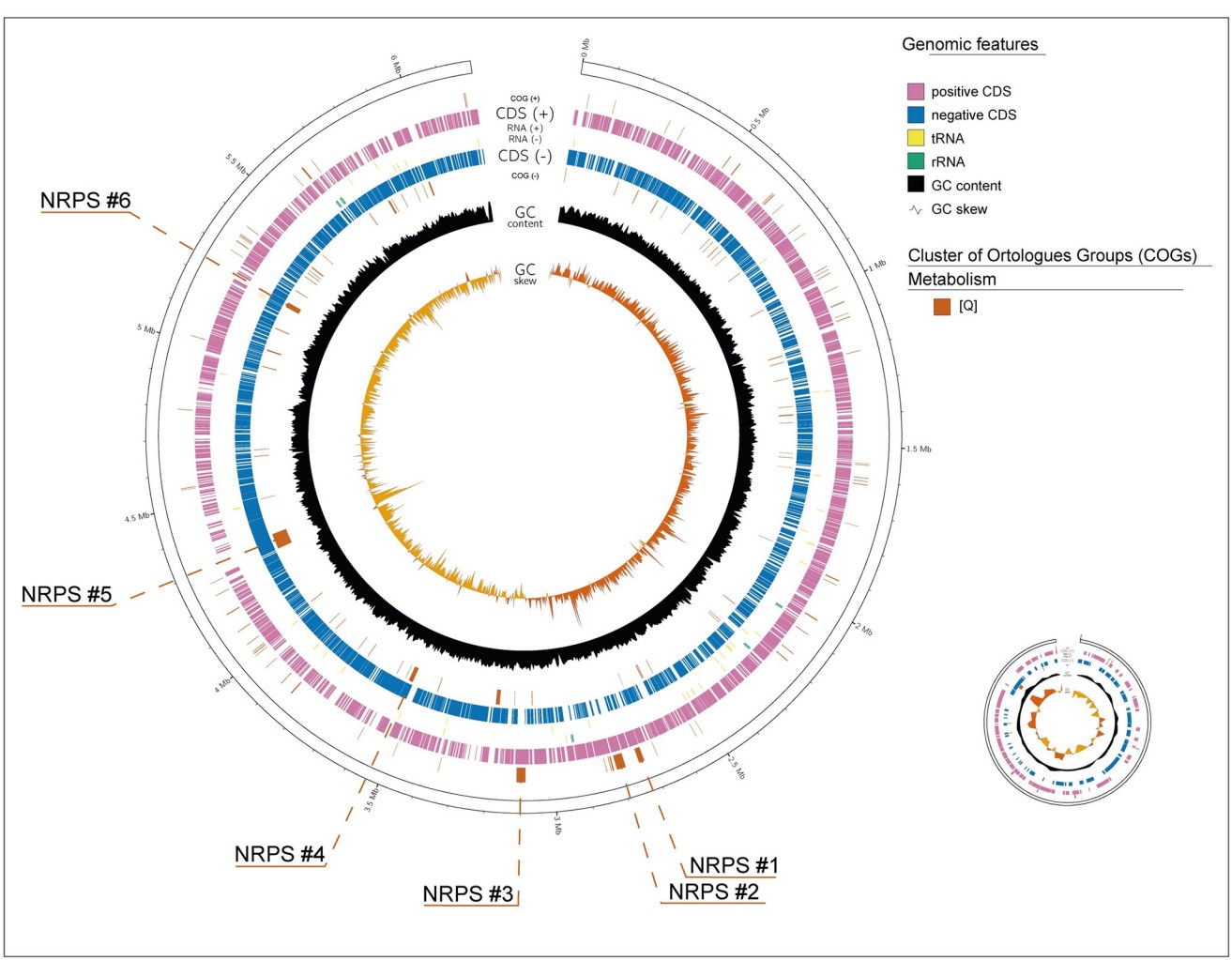

**Fig 4. Visualization of NRPS biosynthetic gene clusters within the genome of *Rhodococcus* sp. H-CA8f.** Genovi representation of *Rhodococcus* sp. H-CA8f complete genome is depicted with an interior break in the ideogram to easily identify each track (*-s complete*, *-cs dawn*,—*cogs Q*, *-te*). Selection of Q orthologs (Secondary metabolites biosynthesis, transport, and metabolism) within the H-CA8f genome includes NPRS CDSs and can be easily seen by the whole genome representation.

The display of specific COG categories can facilitate genome mining of biosynthetic gene clusters (BGCs) containing large CDS. The genome of strain *Rhodococcus* sp. H-CA8f (GCF_002501585.1) comprises two replicons, a chromosome of 6.19 Mb and a plasmid of 301 Kb (Fig 4. [23]). Genome mining analysis of this strain rendered the presence of 17 BCGs, six of which are non-ribosomal peptide synthetases (NRPSs), highly relevant for the production of specialized metabolites, such as antimicrobials [24]. NRPSs are modular multidomain enzymes which have been reported to be the most prevalent classes of BGCs in *Nocardia* [25], as well as in *Rhodococcus* [24,26]. NRPS BGCs in strain H-CA8f range from 55.5 Kbp to 98.4 Kbp [24], where the core biosynthetic genes range from 12.4 Kbp to 26.9 Kbp. The genome of strain H-CA8f was represented displaying only the COG category Q (secondary metabolites biosynthesis;—*cogs* Q; Fig 4). The presence of the six NRPS BGCs from this strain are highlighted and are easily observable, demonstrating that GenoVi could be a useful tool to quickly display the presence of genes that encode for megaenzymes.

## Visualization and analysis of multiple genomes: *Paraburkholderia*

To address the potential of GenoVi for comparative genomics, a directory containing multiple genomes can be given as input. Each file will be analyzed independently, following the normal workflow of the tool. Output tables describing the genomic features and COG information of each genome analyzed can be of great support for comparative genomics studies. When multiple genomes are given as an input, GenoVi processes each file independently and creates tables summarizing the general statistics, COG identification and COG frequency of every genome analyzed into one file.

To better illustrate the usage of the output tables rendered by GenoVi, a genomic analysis was performed on 36 complete genomes of *Paraburkholderia* to identify genomic traits for its replicon classification (S1 Text and S1 Table). *Paraburkholderia* is a bacterial genus encompassing strains often isolated from plant, insect, soil, and anthropogenic-impacted sites [27–29]. Diverse pollutant-degraders and plant-growth-promoting bacteria (PGPB) are part of this taxon, including the degrader of polychlorobiphenyls and aromatic compounds *Paraburkholderia xenovorans* LB400[T] [30], BTEX and hydrocarbon degrader *Paraburkholderia aromaticivorans* BN5[T] [31], and the PGPB model bacteria *Paraburkholderia phytofirmans* PsJn[T] [32].

*Paraburkholderia* is characterized by multipartite genomes comprised of at least two large replicons, a larger element referred to as chromosome or first chromosome in some studies (C1), and a chromid generally designated as the second chromosome (C2). Other genetic elements are usually detected in the genus, including other possible chromids or megaplasmids, classified into each type depending on the presence of indispensable genes for cellular viability. Plasmids are also generally encountered in *Paraburkholderia* genomes [31,33]. Chromosome identification is an easy task, as the genetically stable largest replicon, while chromid—megaplasmid—plasmid categories are not as easily distinguished. Corroboration of core genes, plasmid-type maintenance and replication proteins, codon usage, size, GC-content, and dinucleotide relative abundance distance have been used as markers to classify replicons into chromids, megaplasmids, or plasmids in multipartite genomes [34–36]. Intricate classification boundaries, more than three replicons in most of *Paraburkholderia*, and a wide size range translate into a non-trivial replicon classification in *Paraburkholderia*, leading to misclassifications. While the misidentification of a replicon type does not imply an error for general genomic studies, an easier approach could facilitate the study of the evolutionary relatedness across replicons from the same taxa, the relevance of inter-replicon transcriptional regulation, and the essential nature of any of the replicons within an organism [34,35].

Previous studies have reported differential COG distribution across replicons of *P. xenovorans* LB400[T] and other bacteria, describing functional patterns per replicon [30,33,37,38]. diCenzo *et al*, corroborated a functional bias between each replicon class regardless of their phylogeny, highlighting the presence of transposable elements in plasmids [35]. These data suggest that COG distribution patterns across replicons could be used as an effective tool to quickly identify overall genetic functions, facilitating replicon classification into chromosomes, chromids, megaplasmids, or even plasmids.

## COG percentage distribution differentiates three major groups within *Paraburkholderia* multipartite genomes

A total of 146 replicons ranging from 22 kb to 4.94 Mb, from 36 complete genomes were analyzed by GenoVi (S1 Table) [30–32,39–53]. Hierarchical clustering analysis of COG percentage distribution was evaluated to assess functional bias relevance to replicon classification in *Paraburkholderia* (Fig 5A). COG patterns showed three major groups, where the group designated as "Chromosomes" (n = 36) and "Chromids and megaplasmids" (n = 75) displayed a more

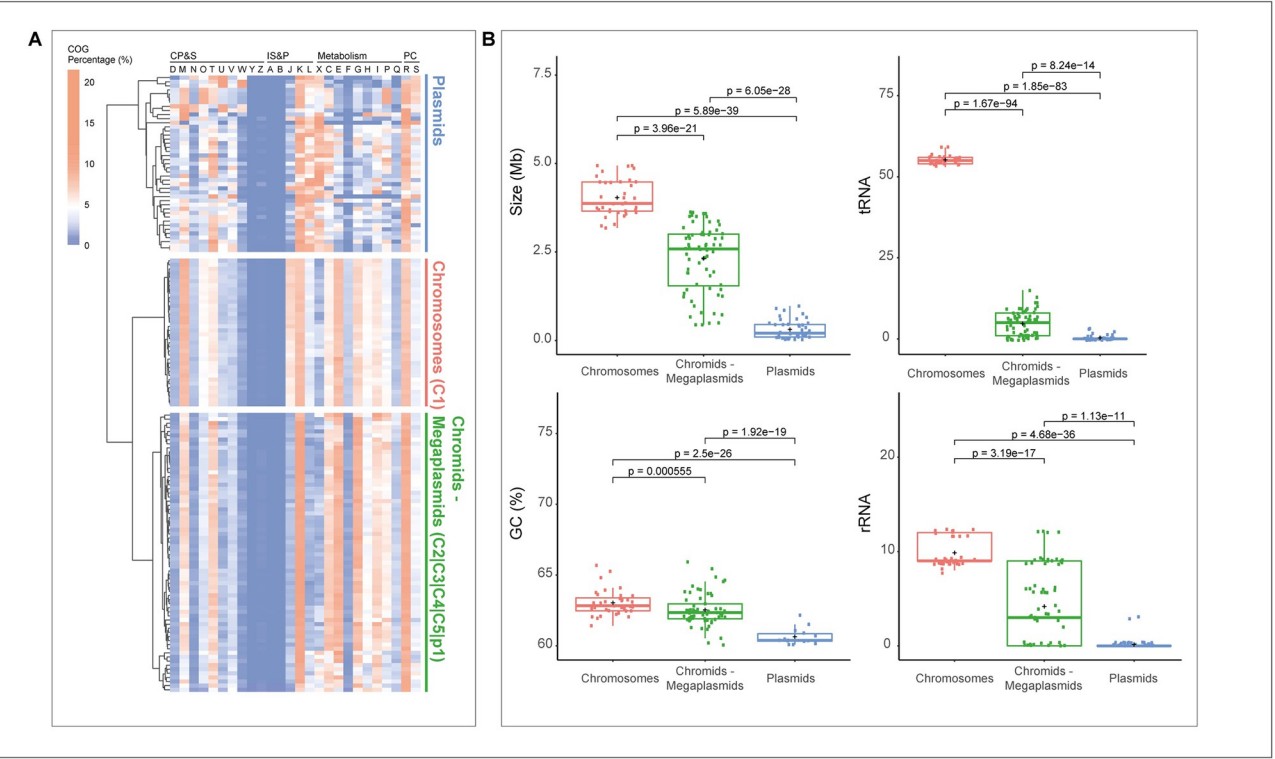

**Fig 5. COG percentage distribution of *Paraburkholderia* replicons elucidates general functional patterns useful for its classification.** (A) COG percentage distribution of the 146 replicons from the *Paraburkholderia* genus. Hierarchical clustering revealed three distinct groups: Major chromosomes C1 (red); Minor chromosomes C2-C3-C4-C5-p1 (green); Plasmids (blue). CP&S: Cellular Processes and Signaling; IS&P: Information Storage and Processing; PC: Poorly Characterized. (B). Genomic features (size, GC-content, tRNA, and rRNA) from each replicon type were identified by COG percentage distribution. tRNA is the best feature to identify chromosomes (C1).

conserved functional pattern, while the class designated as "plasmids" (n = 35) did not possess a clear functional organization besides a general enrichment in the COG class X (Mobilome: prophages, transposons) (Fig 5A and S1 Fig). As expected, only the largest replicon from each genome analyzed is encountered in chromosomes (Fig 5A and S1 Fig). The second larger replicon was always allocated in the chromids and megaplasmid group, alongside other large secondary replicons and megaplasmids (*e.g.*, strain LB400 megaplasmid) confirming the functional bias in *Paraburkholderia* secondary large replicons already observed in other multipartite genomes [35].

Overall genomic features from each replicon type emphasize the segregation of the three groups, observing statistical differences ($p<0.001$) in size, GC-content, tRNAs, and rRNAs among every group (Fig 5B). However, only the presence of a complete tRNAs repertoire (>50) can be used as a specific marker for identification of chromosomes. Other general features, such as size or GC-content can be helpful to guide the identification of chromosomes or plasmids, but some outliers in each group hamper their usage as specific markers.

In general, *Paraburkholderia* chromosomes possess a similar pattern in comparison to chromids and megaplasmids, with only a few COG categories showing significant differences between both groups (Fig 5 and S2 Fig). Regardless, the COG category J has an average representation of 6.08 ± 0.59% in chromosomes, while in chromids-megaplasmids is about 1.92 ± 0.93%, adding another signature feature to distinguish between chromosomes and

other secondary large replicons (Fig 5). Chromids and megaplasmids of *Paraburkholderia* have a higher COG category K, associated with additional transcriptional regulation mechanisms part of larger secondary replicons [35] (S2 Fig).

Discrimination between plasmids and megaplasmids has been delimited by setting an arbitrary boundary size (~350 kb) which may consider megaplasmids as large plasmids that do not follow some other features of this replicon type [35]. While functional distribution cannot identify a similar copy number than chromosomes or an independent partitioning system, it can discern which replicons are distantly related and functionally unstable in relation to the chromosomes and chromids from *Paraburkholderia*, indicating at least remote large and small plasmids in the taxa. For the extent of our analysis, plasmids were defined as replicons distantly related to chromosomes, chromids, and megaplasmids in terms of functional organization independently of their size. The group is composed of sequence elements ranging from 22 kb to 971 kb (Fig 5B), mainly enriched in the new COG category X, including transposases, integrases, and other mobile elements incorporated in the last update of the COG database 2020 [5]. Categories D (Cell cycle control, cell division, chromosome partitioning) and L are also higher in this group than other replicon types, while categories associated with amino acid and coenzyme transport/metabolism (E and H) are less represented in comparison to chromosomes, chromids or megaplasmids (Fig 5A and S2 Fig).

*Paraburkholderia terrae* KU-15 complete genome is displayed as an example to quickly identify and classify each replicon type using general features and functional profiling delivered by GenoVi (Fig 6; [39]). The genome architecture of strain KU-15 is comprised by six replicons, the chromosome (3.74 Mb) and five other replicons of 2.88 Mb, 2.29 Mb, 754.53 kb, 692.03 kb and 64.71 kb (Fig 6A). COG categories are differentially enriched in each replicon displayed by the heatmap obtained by GenoVi. The first replicon (chr1) is the largest, enriched in J orthologs, as commonly seen for other chromosomes (Fig 6B). Replicons chr2 and chr3 possess a higher proportion of K and R categories, a characteristic of chromids and megaplasmids. Chr6 is the smallest replicon, possessing a much higher L and X categories. Interestingly, chr4 and chr5 also exhibited functional patterns usually found on plasmids (Fig 6B).

General features and functional profiling of *Paraburkholderia* replicons can help to quickly guide the identification of chromosomes, chromids, and megaplasmids within this taxon, although thresholds based on experimental validation are needed to set up accurate boundaries for their classification. Nevertheless, the development of bioinformatic tools that help researchers to easily obtain genome profiling data allows us to identify patterns shedding novel evidence about genome evolution, organization, and architecture within bacterial and archaeal genomes.

GenoVi uses a single input file and creates a customizable circular genomic map in one step, including a built-in COG categories analysis by alignment free-methods, generating a scaled circular representation for complete genomes or multiple replicons, and, therefore, delivering genomic data for comparative genomics analyses, and ready-to-publish circular representations.

## Conclusion

GenoVi is an open-source and easy-to-use Python command-line application for the creation of custom circular genome representations of complete and draft genomes, and multiple replicons of bacteria and archaea. It allows COG categories analysis via alignment-free methods, and automatic scaling for complete genomes, which provide the easy visualization of genomic features. Genomic features and COG distribution patterns obtained by GenoVi, are a useful

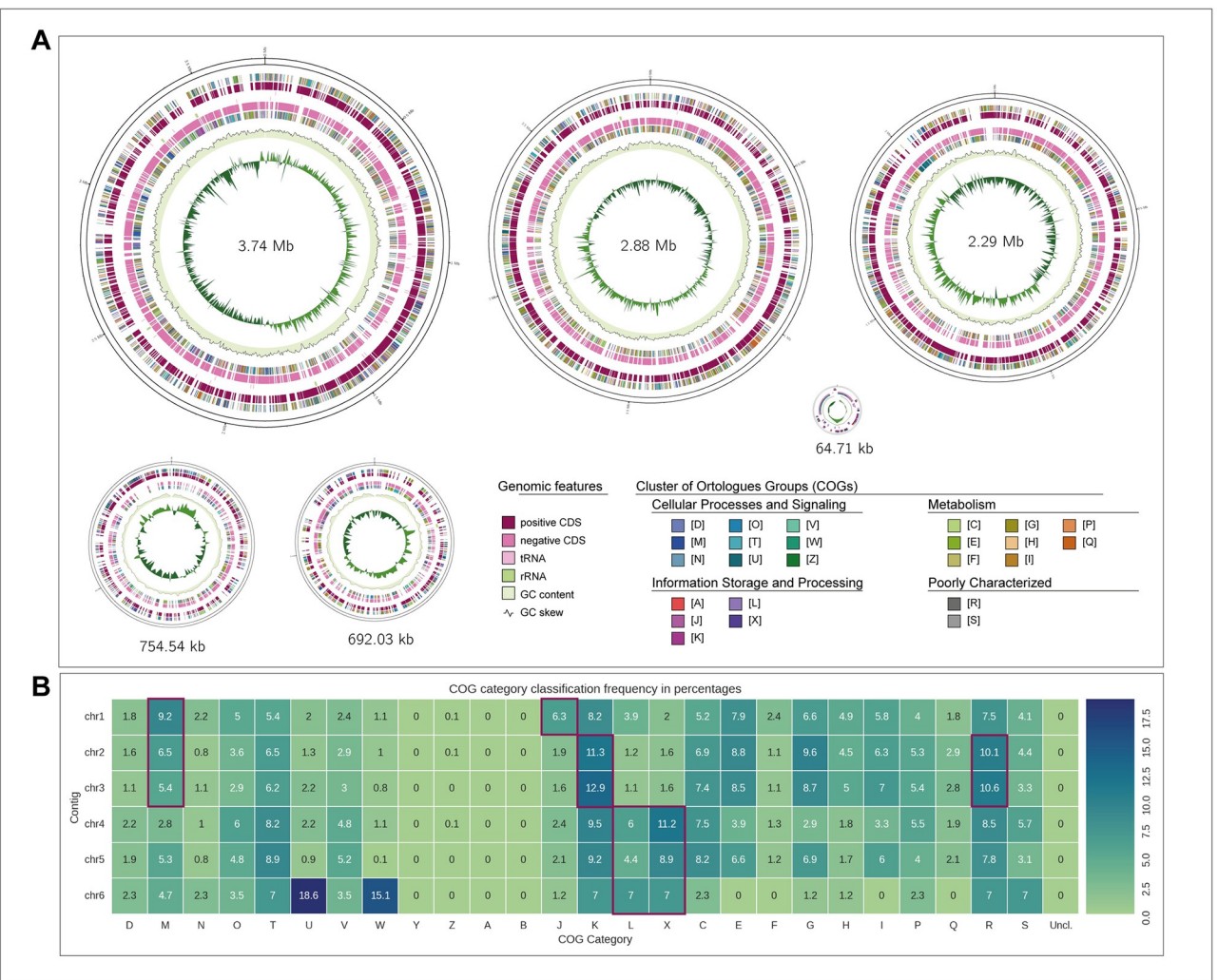

**Fig 6.** *Paraburkholderia terrae* **KU-15 visualization and analysis by GenoVi allows its replicon classification.** (A) Complete genome representation of *P. terrae* KU-15 using the blossom palette (-*cs blossom*). (B) Heatmap generated by GenoVi representing COG percentage frequency of each replicon of *P. terrae* KU-15 genome. Red boxes highlight key COG categories to differentiate each replicon type.

method to quickly discriminate between replicon types in multipartite genomes, as corroborated in the complete genomes from *Paraburkholderia*.

## Availability and Future Directions

GenoVi is freely available under a BY-NC-SA Creative Commons License and can be downloaded from https://github.com/robotoD/GenoVi. GenoVi can be obtained in two steps: Creating a Conda environment with Circos, followed by installation using the package-management system pip with pip install genovi. Also, a Docker container of GenoVi is available. Genomes used in this study are available at https://zenodo.org/record/7331473. The software is open and we expect researchers to implement it in their routine genomic analyses, being able to request new features that could be implemented. Nevertheless, our team is invested in delivering an interactive web-platform to improve usability for users in biological sciences, which can easily analyze and visualize MAGs or genomic data of single microorganisms,

including new modules to highlight a specific loci or locus in the sequence, or adding extra annotation tools that could be useful for environmental or clinical fields.

## Supporting information

**S1 Fig. Functional distribution patterns show three major groups of the 147 replicons from the *Paraburkholderia* genus.** Hierarchical clustering of COG percentage shows three major groups in *Paraburkholderia* replicons.
(TIF)

**S2 Fig. Functional distribution per replicon type represented by COG percentage.**
(TIF)

**S1 Table. General features of *Paraburkholderia* genomes used in this study.**
(PDF)

**S1 Text. Supplementary methods.**
(DOCX)

## Acknowledgments

We kindly thank Leonardo Zamora for his contribution to software validation.

## Author Contributions

**Conceptualization:** Andrés Cumsille, Roberto E. Durán, Mauricio Araya, Nicolás Jara, Carlos Buil-Aranda.

**Data curation:** Andrés Cumsille, Roberto E. Durán, Andrea Rodríguez-Delherbe, Vicente Saona-Urmeneta.

**Formal analysis:** Andrés Cumsille, Roberto E. Durán, Andrea Rodríguez-Delherbe, Vicente Saona-Urmeneta.

**Funding acquisition:** Andrés Cumsille, Roberto E. Durán, Michael Seeger, Mauricio Araya, Nicolás Jara, Carlos Buil-Aranda.

**Investigation:** Andrés Cumsille, Roberto E. Durán, Andrea Rodríguez-Delherbe, Vicente Saona-Urmeneta.

**Methodology:** Andrés Cumsille, Roberto E. Durán, Andrea Rodríguez-Delherbe.

**Project administration:** Roberto E. Durán, Carlos Buil-Aranda.

**Resources:** Roberto E. Durán, Beatriz Cámara, Michael Seeger, Carlos Buil-Aranda.

**Software:** Andrés Cumsille, Roberto E. Durán, Andrea Rodríguez-Delherbe, Vicente Saona-Urmeneta.

**Supervision:** Roberto E. Durán, Carlos Buil-Aranda.

**Validation:** Andrés Cumsille, Roberto E. Durán, Andrea Rodríguez-Delherbe, Vicente Saona-Urmeneta.

**Visualization:** Andrés Cumsille, Roberto E. Durán, Andrea Rodríguez-Delherbe, Vicente Saona-Urmeneta.

**Writing – original draft:** Andrés Cumsille, Roberto E. Durán.

**Writing – review & editing:** Andrés Cumsille, Roberto E. Durán, Andrea Rodríguez-Del-herbe, Vicente Saona-Urmeneta, Beatriz Cámara, Michael Seeger, Mauricio Araya, Nicolás Jara, Carlos Buil-Aranda.

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
