## [Decision Letter · Decision Letter 0]

15 Oct 2022

Dear Mr. Durán,

Thank you very much for submitting your manuscript "GenoVi, an open-source automated circular genome visualizer for bacteria and archaea uncovers genomic patterns for replicon identification in multipartite genomes" for consideration at PLOS Computational Biology.

As with all papers reviewed by the journal, your manuscript was reviewed by members of the editorial board and by several independent reviewers. In light of the reviews (below this email), we would like to invite the resubmission of a significantly-revised version that takes into account the reviewers' comments.

The reviewers generally agree that the tool is useful and the manuscript is well written, but concerns remain. One important question (raised explicitly by reviewer #1, but also perhaps underlying some of the concerns of reviewer #3) is whether the example of Paraburkholderia (which covers most of the figures as well as the results section) is a good choice to demonstrate the value of GenoVi. For example, Fig. 3 does not seem particularly informative. The subsequent figures seem much clearer, but they also seem (as per the comment of reviewer #3) to have been generated by a different programme. While we don't find the use of companion tools to be objectionable per se (in fact, it can be salutary to not reinvent the wheel), in this case, the added value of GenoVi is unclear.

We cannot make any decision about publication until we have seen the revised manuscript and your response to the reviewers' comments. Your revised manuscript is also likely to be sent to reviewers for further evaluation.

Sincerely,

Luis Pedro Coelho

Academic Editor

PLOS Computational Biology

Lucy Houghton

Staff

PLOS Computational Biology

The reviewers generally agree that the tool is useful and the manuscript is well written, but concerns remain. One important question (raised explicitly by reviewer #1, but also perhaps underlying some of the concerns of reviewer #3) is whether the example of Paraburkholderia (which covers most of the figures as well as the results section) is a good choice to demonstrate the value of GenoVi. For example, Fig. 3 does not seem particularly informative. The subsequent figures seem much clearer, but they also seem (as per the comment of reviewer #3) to have been generated by a different programme. While we don't find the use of companion tools to be objectionable per se (in fact, it can be salutary to not reinvent the wheel), in this case, the added value of GenoVi is unclear.

Reviewer's Responses to Questions

**Comments to the Authors:**

Reviewer #1: The manuscript describes a command line bioinformatics tool that is designed to take a genbank file and generate a circos plot to display the GC content and skew, as well as ortholog positions using the COGs database. Creating circos plots with the circos program is a non-trivial task with a steep learning curve. The advantage of this tool is in the automated processing and rendering of circos plots to display information that the authors believe would be most relevant to users when visualising bacterial genomes. There are other platforms and pipelines designed to help users create circos plots, but each have their own strengths and weaknesses and having more options available to end users is always a good thing. The manuscript is well written, and the tool performs well.

Main concerns:

GenoVi is sold as a visualisation tool, and an example is given for how GenoVi can help distinguish replicons from plasmids in a bacterial genus. The example uses the COG annotations generated by GenoVi as input for a gene composition-based analysis, but it does not indicate whether or not the GenoVi visualisations themselves were at all helpful in this endeavour. My impression is that the findings of the example could be achieved by simply annotating the genome with DeepNOG and then preforming the subsequent analyses presented in the results. It would be beneficial to better highlight the usefulness of the visualisations that GenoVi creates.

Figure 2

It’s impossible to tell what each of the tracks are. There are too many colours to simply rely on the colours alone to know what is what. I suggest using ideogram spacing and adding in labels for each track. http://www.circos.ca/documentation/tutorials/ideograms/spacing_breaks/lesson

Minor comments and suggestions:

For draft genomes, how does GenoVi lay out the contigs (order and orientation) and can this interfere with GC skew calculations at the boundaries?

For draft genomes, my understanding is that plasmid contigs and genome contigs are combined in the same circos plot. Is this correct? While this might not be ideal, there probably isn’t a neat or easy solution for this. Maybe it would be best to just make sure this is clear in the documentation.

“””Interactive web platform””” – this would be a great future addition.

“””GenoVi is the first open-source tool that uses a unique input file …”””

Genbank files are not unique, which is good because you don’t want unique (non-standard) file formats for your inputs.

Running the Program:

Do you need bioconda and conda-forge channels when installing circos?

Are there plans to add this to bioconda? This would make the installation of GenoVI and Circos a one-step process.

As the input file is always required, it could just be a bareword argument, e.g.

genovi input.gb …

Instead of specifying -status every time, consider having a -draft flag (and default to complete).

e.g. for complete genome:

genovi input.gb

e.g. for draft genome:

genovi input.gb -draft

I would call the output folder name “genovi”, rather than the name of someone else’s program.

consider support for gzipped genbank files

Reviewer #2: The authors have developed Python command-line application for the visualization circular genome representations of complete and draft genomes of bacteria and archaea. However, there are scope to impoverish the manuscript before acceptance.

Therefore, I recommend major revision as per the following suggestions.

1. Authors should include a comparison table highlighting advantages of their new tool over already exiting tools.

2. I suggest to containerize their software using docker or singularity for portability across machine.

Reviewer #3: It looks like the tool is useful for automatically creating circular genomics visualization. However, biggest concern I have is that the resulting circular visualizations themselves do not seem to enable complete analysis, and many parts of the analysis in the Results and discussion section actually seem to rely on externally-generated visualization, such as heatmaps and box plots.

- I wonder if the heatmaps and box plots illustrated in the manuscript can be generated automatically along with the circos plots. I think this will make the analysis process more seamless to users.

- One of the main limitation of the circular visualizations in term of their interpretation is that it is difficult to see the COG distribution patterns accurately due to the use of large number of similar colors. For example, in Fig. 2A, it is really hard to find what are the most representative COG categories. I wonder if it makes sense to enable users to filter COG categories, show only top-N categories, or etc.

- Many color palettes seem to be not color-blindness friendly. I suggest using color-blindness friendly palette as a default one (e.g., https://colorbrewer2.org/ or https://mikemol.github.io/technique/colorblind/2018/02/11/color-safe-palette.html).

- Sufficiently distinct colors for individual categories should be used whenever possible (e.g., currently, positive CDS, negative CDS, and rRNA are too similar which is difficult to differentiate in the visualization).

- Using the "complete" sequence option for the first tutorial example gives an error on my machine (i.e. "FileNotFoundError: [Errno 2] No such file or directory: 'circos-contig_1.png' -> 'circos/circos-contig_1.png'").

- Having more test datasets in the repository would be valuable.

Minor

- Color legends generated are sometimes too small to read.

- In general, I think interactive visualizations can make the analysis more effective (e.g., using Gos Python package, https://osf.io/yn3ce/), such as using mouse hover to see exact COG categories in tooltips.

**Have the authors made all data and (if applicable) computational code underlying the findings in their manuscript fully available?**

Reviewer #1: Yes

Reviewer #2: Yes

Reviewer #3: Yes

PLOS authors have the option to publish the peer review history of their article (what does this mean?). If published, this will include your full peer review and any attached files.

Reviewer #1: **Yes: **Michael J. Roach

Reviewer #2: **Yes: **Vikash K Singh

Reviewer #3: No
---

## [Decision Letter · Decision Letter 1]

20 Jan 2023

Dear Mr. Durán,

Thank you very much for submitting your manuscript "GenoVi, an open-source automated circular genome visualizer for bacteria and archaea" for consideration at PLOS Computational Biology. As with all papers reviewed by the journal, your manuscript was reviewed by members of the editorial board and by several independent reviewers. The reviewers appreciated the attention to an important topic. Based on the reviews, we are likely to accept this manuscript for publication, providing that you modify the manuscript according to the review recommendations.

We are in principle happy to accept the manuscript and ask only that the authors fix the small textual mistake pointed out by reviewer #1

Sincerely,

Luis Pedro Coelho

Academic Editor

PLOS Computational Biology

Lucy Houghton

Staff

PLOS Computational Biology

We are in principle happy to accept the manuscript and ask only that the authors fix the small textual mistake pointed out by reviewer #1

Reviewer's Responses to Questions

**Comments to the Authors:**

Reviewer #1: The authors have addressed our main concerns. The updated GitHub and user guide looks great!

Error in the Author summary: "difficulting its usability"

Ideas for future developments:

Consider support for alternative annotations to COGs, e.g. KEGG, Phrogs/pvogs for viral annotations etc.

Test dataset for Archaea

Reviewer #2: The authors have taken into account the previous suggestions and remarks. The present manuscript is suitable for publication.

Reviewer #3: Thank you for making changes. I do not have further comments.

**Have the authors made all data and (if applicable) computational code underlying the findings in their manuscript fully available?**

Reviewer #1: Yes

Reviewer #2: Yes

Reviewer #3: Yes

PLOS authors have the option to publish the peer review history of their article (what does this mean?). If published, this will include your full peer review and any attached files.

Reviewer #1: **Yes: **Michael Roach

Reviewer #2: **Yes: **Vikash Kumar Singh

Reviewer #3: **Yes: **Sehi L'Yi

Figure Files:

Data Requirements:

Reproducibility:

References:

---

## [Editor Report · Decision Letter 2]

5 Mar 2023

Dear Mr. Durán,

We are pleased to inform you that your manuscript 'GenoVi, an open-source automated circular genome visualizer for bacteria and archaea' has been provisionally accepted for publication in PLOS Computational Biology.

Best regards,

Luis Pedro Coelho

Academic Editor

PLOS Computational Biology

Lucy Houghton

Staff

PLOS Computational Biology

---

## [Editor Report · Acceptance letter]

30 Mar 2023

PCOMPBIOL-D-22-01291R2 

GenoVi, an open-source automated circular genome visualizer for bacteria and archaea

Dear Dr Durán,

I am pleased to inform you that your manuscript has been formally accepted for publication in PLOS Computational Biology. Your manuscript is now with our production department and you will be notified of the publication date in due course.

With kind regards,

Anita Estes
